# Genotypic Difference in the Responses to Nitrogen Fertilizer Form in Tibetan Wild and Cultivated Barley

**DOI:** 10.3390/plants10030595

**Published:** 2021-03-22

**Authors:** Shama Naz, Qiufang Shen, Jonas Lwalaba Wa Lwalaba, Guoping Zhang

**Affiliations:** Department of Agronomy, Key Laboratory of Crop Germplasm Resource of Zhejiang Province, Zhejiang University, Hangzhou 310058, China; 11716100@zju.edu.cn (S.N); shenqf@zju.edu.cn (Q.S.); jonaslwalaba@zju.edu.cn (J.L.W.L.)

**Keywords:** barley, biomass, genotype, nitrogen fertilizer, photosynthesis

## Abstract

Nitrogen (N) availability and form have a dramatic effect on N uptake and assimilation in plants, affecting growth and development. In the previous studies, we found great differences in low-N tolerance between Tibetan wild barley accessions and cultivated barley varieties. We hypothesized that there are different responses to N forms between the two kinds of barleys. Accordingly, this study was carried out to determine the response of four barley genotypes (two wild, XZ16 and XZ179; and two cultivated, ZD9 andHua30) under 4Nforms (NO_3_^−^, NH_4_^+^, urea and glycine). The results showed significant reduction in growth parameters such as root/shoot length and biomass, as well as photosynthesis parameters and total soluble protein content under glycine treatment relative to other N treatments, for both wild and cultivated barley, however, XZ179 was least affected. Similarly, ammonium adversely affected growth parameters in both wild and cultivated barleys, with XZ179 being severely affected. On the other hand, both wild and cultivated genotypes showed higher biomass, net photosynthetic rate, chlorophyll and protein in NO_3_^−^ treatment relative to other three N treatments. It may be concluded that barley undisputedly grows well under inorganic nitrogen (NO_3_^−^), however in response to the organic N wild barley prefer glycine more than cultivated barely.

## 1. Introduction

Nitrogen (N) plays fundamental roles in plant growth and development, as it is a necessary component of many biological macromolecules, including proteins, nucleic acids and hormones [1]. To maintain healthy growth and development, plants require sufficient N from soils [2]. N deficiency can severely affect plant growth and development by disrupting important biological processes such as protein synthesis and photosynthesis [3,4], reflected by reduced leaf area, plant height, and biomass accumulation [5]. On the other hand, excessive N application causes an adverse effect on plant growth and resistance to abiotic/biotic stresses [6,7], and also results in severe environmental pollution [8]. Therefore, it is quite important to improve N use efficiency of crops [9].

Plants can use many kinds of N forms, ranging from inorganic N such as NH_4_^+^ and NO_3_^−^ to polymeric N such as proteins [10]. The uptake of different organic and inorganic N forms varies with plant species and availability of various N forms in soils [11]. For example, rice and conifer grew better under NH_4_^+^ nutrition [12,13], while most crops, including maize, wheat, bean, eggplant, cucumber, tomato and barley prefer nitrate N [14,15,16,17,18]. Cucumber plants grown under NH_4_^+^-nutrition showed slower growth in comparison with those grown under NO_3_^−^ nutrition [19]. NH_4_^+^ nutrition has been generally considered as toxic for plants, particularly when NH_4_^+^ is supplied as a sole N source [20]. For some plants, at low concentrations (<3 mM), NH_4_^+^ is typically preferred N source, but above a certain threshold, NH_4_^+^ becomes toxic [20]. This threshold varies with plant species and genotypes within a species as well as environmental factors [20,21]. Furthermore, it was reported that plants not only utilize inorganic N, but also can absorb many organic N [22,23]. Glycine is the most commonly used amino acid in plants because of its simplicity, low molecular weight and ratio of carbon and nitrogen, and rapid diffusion rate in soil [23]. Wang et al. (2014) found the differences in physiological and proteomic response to glycine nutrition between two pakchoi cultivars [24]. In wheat and *Arabidopsis*, Rubio-Asensio and Bloom (2017) found that the elevated atmospheric CO_2_ had the negative impact on plant N status, enhancing the dependence on N form [25]. In short, the results in studies on the responses of crops to the different forms of N fertilizers are controversy up to date.

Barley (*Hordeum vulgare*) is known as one of the earliest domesticated cereal crops, and there continues to be a rise in the interests associated with its benefits as a major malting material and as a source of healthy food [26]. In barley production, nitrogen is the crucial nutrient input for achieving high yield, and this crop is susceptible to insufficient N supply. Tibetan annual wild barley (*Hordeum spontaneum*) is considered as an ancestor of the modern cultivated barley and is rich in genetic diversity [27]. In the previous studies, we identified some Tibetan wild barley accessions with high low N tolerance [28], and also found that higher low N tolerance in the wild barley compared to the cultivated barley could be attributed to larger N uptake ability by roots and better N utilization in shoots, which are in turn associated with higher expression of nitrate transporters and energy-saving N assimilation pattern [29,30,31]. The comparative studies of Tibetan wild barley and cultivated barley in low N tolerance showed the exact presence of the specific mechanisms in N assimilation for the wild barley. Thus, we hypothesize that there may be difference in the response to the different N forms between the two barleys (wild and cultivated). Accordingly, this study was conducted to understand the influence of different nitrogen form (nitrate, ammonium, urea, and glycine) on growth and N assimilation of both wild and cultivated barley.

## 2. Results

### 2.1. Plant Growth, Root and Shoot Dry Weight

The influence of N forms on plant morphology (Figure 1A–D) and growth parameters (Figure 1E–J) of the four cultivars are presented in Figure 1. A significant difference was found among the four N forms and four barley cultivars. For XZ16, urea and glycine treatments had the highest and lowest plant height, respectively (Figure 1E). For XZ179, ammonium treatment had the lowest plant height, being significantly lower than other three treatments. For ZD9 and Hua30, nitrate and glycine treatments had the highest and lowest plant height, respectively.

Root length was significantly shorter in the glycine treatment than in other treatments (Figure 1F). In the ammonium treatment, ZD9 and XZ179 had significantly lower root length thanXZ16 and Hua30. In general, XZ16 and Hua30 did not show a significant reduction under ammonium in comparison with other N treatments. Significant difference was found among barley genotypes and N forms in root and shoot weight. The plants treated with urea had the largest biomass for all genotypes, being significant difference with those treated by other three nitrogen forms, except XZ179 which had the smallest biomass under ammonium treatment (Figure 1G,H). In addition, XZ16 and ZX179 had fewer tillers per plant in ammonium treatment than in nitrate treatment, while ZD9 and Hua30 did not show any difference between the two N treatments (Figure 1I,J).

### 2.2. Root Architecture

The influence of N forms on root architecture of four barley genotypes is presented in Figure 2. Except XZ179, other three genotypes had significantly shorter total root length in glycine treatments relative to other three N forms (Figure 2A). For total root area, all the genotypes except XZ179 had lower values in glycine treatment. Similarly, ammonium reduced root area for all genotypes except ZD9, which had no significant difference between nitrate and ammonium treatments (Figure 2B). For secondary roots, largest number was observed in urea treatment for all genotypes, except Hua30 which showed the largest number under nitrate treatment, while the lowest value was recorded under glycine treatment for all genotypes except XZ179, which showed the lowest value in ammonium treatment (Figure 2C). Root diameter was smaller in urea treatment than in other treatments for all genotypes (Figure 2D). Except XZ179, all the genotypes had the smallest root volume and root tips per plant under glycine treatment (Figure 2E,F).

### 2.3. Photosynthetic Parameters, Chlorophyll and Carotenoid Content

N forms had significant effect on all photosynthetic parameters of four barley genotypes (Figure 3). Photosynthetic rate (Pn) was highest and lowest in nitrate and glycine treatments, respectively for all genotypes (Figure 3A). The lower *Gs* and Tr values were found in glycine treatment (Figure 3B,C). For transpiration rate (Tr), the highest value occurred in urea treatment for XZ16 and in nitrate treatment for other three genotypes (Figure 3D). Interestingly, the reverse pattern was noted for Ci value, which had the highest values in glycine treatment for all genotypes.

Chlorophyll content was significantly lower in glycine treatment than that in other three treatments for all genotypes (Figure 4A,B,D), with urea treatment having the maximum value. Comparatively, the two wild barley accessions had relatively higher chlorophyll content than the two cultivated barley genotypes in glycine treatment. For XZ16 and Hua30, carotenoid content was lower under urea as compared to other treatments, while for XZ179 and ZD9 lowest carotenoid could be seen under ammonium and nitrate treatment, respectively (Figure 4C).

### 2.4. Tissue Nitrogen and Soluble Protein Concentrations

The influence of four N treatments on total N concentration in roots and shoots of four barley genotypes is presented in Figure 5. Shoot N concentration of XZ16, XZ179 and Hua30 did not show significant difference among four N treatments (Figure 5A), while ZD9 had significantly higher N concentration in nitrate and glycine treatments than in other two treatments. Similarly, no significant difference was found in root N concentration among all N treatments for all genotypes, except Hua30 which had significantly higher root N concentration in nitrate than in other treatments (Figure 5B).

For shoot soluble protein (SP) concentration, glycine treatment had lower value while nitrate treatment had the highest value for all genotypes (Figure 5C). Ammonium also reduced total soluble proteins in shoots of all genotypes except XZ16 relative to other two treatments (nitrate and urea). Likewise, glycine and nitrate treatments reduced and increased root soluble proteins in all genotypes, respectively, in comparison with other treatments (Figure 5D).

## 3. Discussion

In this study we observed the difference between wild and cultivated barley in their responses to organic N (glycine). Ammonium appears to be a toxic N source for barley growth as reported earlier in cucumber [19]. In comparison with XZ16, XZ179 had more reduction in growth under ammonium treatment relative to nitrate treatment. XZ16 was also reported tolerant to many stresses such as drought, salinity, and aluminum [27,32]. ZD9 and Hua30 are all characterized by high grain yield, although they differ greatly in yield components, with ZD9 having more grains per spike and larger kernel, and Hua30 having more spikes per plant and fewer grains per spike. It is interesting that the two cultivars showed the different responses to N forms in terms of growth and physiological parameters. For ZD9 higher shoot N concentrations could be found in nitrate and glycine treatments, while for Hua30 higher root N concentration occurred in the nitrate treatment. However, the mechanisms for these differences in N form response need to be clarified. Many studies confirmed that plants were also able to use organic N, including amino acids, peptides and proteins [23]. In this study the plants subjected to glycine showed N starvation symptoms, characterized by decreased total biomass and root/shoot length for all genotypes, which is in agreement with the previous studies [33,34]. Similarly, in lettuce, fresh and dry weight were significantly reduced under glycine-N supply as compared to nitrate N [35]. Wang et al. (2014) reported the different responses of two pakchoi cultivars to glycine and attributed the difference between the two cultivars to up or down-regulation of certain glycine responsive proteins associated with plant defense or stress energy and N metabolism [24].

We found greater total root length, root volume, and root surface area in nitrate treatment than in ammonium treatment for all genotypes (Figure 2), which might be attributed to the fact that for barley NH_4_^+^ assimilation demands more assimilates, thus resulting in less biomass accumulation [36]. In addition, glycine treatment had less root growth in all genotypes except XZ179. The same behavior was also noticed in various barley genotypes by other authors [37,38] as well as in other plant species [39,40] under low N stress. Domínguez-May et al. (2013) also found that glutamate and aspartate did not inhibit the root growth in pepper while glycine inhibited root growth [41]. In the current study, stunt root growth was observed in ammonium treatment, especially in XZ179, which was considered as the major symptom of ion toxicity for barley [13,42].It can be seen from Figure 1 that roots of the two wild genotypes were more seriously affected than shoots. It was suggested that the central part accountable for ammonium toxicity is root where mostly NH_4_^+^ metabolism takes place [43].

Carotenoids have protective role, protecting chlorophyll from photo-oxidation under stress conditions, thus the genotypes with higher carotenoid content should be favorable for fighting abiotic stress [44]. Similarly, we also noted that under glycine carotenoids content increased while total chlorophyll content decreased relative to other treatments. Ali et al. (2013) reported that the Ca(NO_3_)_2_ fed plants had less oxidative stress than the plants fed with other two N forms (urea or (NH_4_)_2_SO_4_),reflected by the higher activities of the antioxidative enzymes and the higher content of the non-enzymatic antioxidants (carotenoids) in these plants [45].

In this study we did not find any significant difference in N concentration of both shoots and roots among four N treatments (Figure 5A,B), however there was a significant difference in total soluble proteins among treatments and genotypes (Figure 5C,D). External N affects free amino acids and proteins metabolism, thus resulting in changes in N uptake, transport and metabolisms. It was reported that free amino acids content in plant tissues was greatly affected by N forms [46]. In this study, glycine treatment induced obvious low N stress for all genotypes, as reflected by lower Total soluble proteins (TSP) in both shoots and roots. It is well documented that low TSP level means limited protein synthesis [47]. The fine regulation of photosynthetic metabolism is required to adapt to different N source, as photosynthesis is one of the key processes closely related to plant growth [48]. Ammonium treatment decreased stomatal conductance and transpiration in French beans [49] and tobacco [3]. In this study, we also found a significant effect of N forms on photosynthesis in all genotypes (Figure 3). In ammonium treatment photosynthetic rate, stomatal conductance and transpiration were much lower than other N treatments for all genotypes. However, we observed Ci value was relatively greater for all genotypes in glycine treatment. The possible reason might be a result of lower N availability for photosynthesis in this treatment, which caused larger mesophyll cell resistance, thus leading to high CO_2_ concentration in the sub-stomatal cavity of leaf [50]. Similar findings were once found in rice and sunflower [51,52].

## 4. Materials and Methods

### 4.1. Plant Growth Conditions and Experimental Design

A hydroponic experiment was conducted in a greenhouse at Zijingang campus, Zhejiang University, Hangzhou, China. Four barley genotypes, namely XZ16, XZ179 (Tibetan annual wild barley), ZD9 and Hua30 (cultivated) were used. XZ16 was much higher than XZ179 in low N tolerance [29]. ZD9 is a newly released cultivar, while Hus30 is widely planted locally, released more than 20 years ago. ZD9 and Hua30 differ largely in yield components, with Hua30 having more tillers and spikes per plant and ZD9 being greater in kernel weight. Healthy seeds were disinfected with 3% hydrogen peroxide (H_2_O_2_) for 20 min, then rinsed five times in sterile de-ionized water, and soaked in ddH_2_O for three hours. After soaking, the seeds were put in a sand bed and kept for 24 h at 4 °C to break dormancy, and then germinated in a growth chamber. At two-leaf stage, theuniform seedlings were selected and transferred to 5 L pots containing basic nutrient solution on 24 November 2018.

When barley seedlings were at four-leaf stage (stage 2 and day 14 on Biologische Bundesanstalt, Bundessortenamt and Chemical industry scale BBCH scale), N fertilizer treatments were initiated. There were four N fertilizer forms, i.e., nitrate (NO_3_^−^), ammonium (NH_4_^+^), urea and organic N, provided by KNO_3_, (NH_4_)_2_SO_4_, urea and glycine, respectively. The final N concentration of the four N treatments in the nutrient solution was 2 mM and other nutrient concentrations were the same for all four treatments (Appendix A). The experiment was arranged in a split plot design with four replicates, and N form as main plot and barley genotypes as sub-plot. The pH of the solution was adjusted to 5.8 ± 1 with HCl or NaOH as required, and the nutrient solution was continuously aerated and renewed every four days.

### 4.2. Measurement of Morphological Parameters

At the 30th day after treatments (day 44 on BBCH scale), growth parameters such as plant height, root length, leaf area and tillers per plant were measured manually, and then the sampled plants were separated into roots and shoots and dried in an oven with 70 °C for 72 h, and dry weight was recorded. At 28 d after treatments (stage 4 and day 42 on BBCH scale), root morphological parameters, including root total length (cm), root total surface area (cm^2^), number of secondary roots, root diameter (mm), volume (cm^3^) and number of root tips were determined by using WinRhizo Pro (S) v. 2009a software (Regent Instruments Inc., Quebec City, QC, Canada) after scanning with a root scanning machine (Epson, Nagano, Japan) Expression 10000XL with transparency adapter; greyscale, 600 dpi).

### 4.3. Gas Exchange Analysis

Gas exchange parameters including net photosynthetic rate (Pn), stomatal conductance (Gs), intercellular carbon dioxide (CO_2_) concentration (Ci), and transpiration rate (Tr) were measured on the second top most fully expanded leaf using an infra-red gas analyzer (LI-COR 6400, Lincoln, NE, USA) at 28 d after treatments (at 42 day on BBCH scale). All these measurements were carried out during 9.00–12.00 am on the same clear and sunny day. These measurements were conducted in an open system in which the instrument used reference of CO_2_ present in the atmosphere. Five measurements were made for each treatment.

### 4.4. Chlorophyll and Carotenoids Content Determination

Total chlorophyll and carotenoid content was determined using the second top most fully expanded leaf at 29 d after treatments (stage 4 and day 43 on BBCH scale) according to Arnon (1949) [53]. The fresh leaves were chopped into small pieces and suspended into the 80% acetone solution for 12 h. After complete discoloration of the green leaves, the aliquots were centrifuged for 5 min at 10,000× *g*. Chlorophyll a, chlorophyll b and carotenoids were measured at 645, 663 and 480 nm respectively, using 80% acetone as a blank with a spectrophotometer (SPECTROstar Nano, Ortenberg, Germany).

### 4.5. Analysis of Nitrogen Concentration and Total Soluble Protein

Total N concentration in plant tissues (shoot and root) was determined by the Kjeldahl method. At 30d after treatments (stage 4 and day 44 on BBCH scale), both leaf and root samples were collected and dried in an oven at 80 °C for three days. After drying, the plant tissues were ground into a fine powder. About 0.2 g sample powder was digested in sulfuric acid, and analyzed for N content according to Li et al. (2006) [17]. TSP was extracted from the roots and shoots according to Karimzadeh et al. (2006) [54] after 29 days of treatment (stage 4 and day 43 on BBCH scale). Accordingly, the root and shoot were frozen in liquid nitrogen and crushed with a tissue homogenizer. The crushed roots and shoots were dissolved in pre-cooled phosphate buffer with pH 7.8 and made the final volume to 1000 mL. The concentration of protein extracts was determined by a colorimetric method as described by Bradford (1976) [55] using protein assay dye, Coomassie Brilliant Blue G-250. The absorbance was determined at 595 nm. Four independent biological replications were used for each measurement.

### 4.6. Statistical Analysis

Data obtained were analyzed for two-way ANOVA and tested for significant difference using statistical software package (SPSS 19.0, Chicago, IL, USA). As the main objective was to determine the influence of N fertilizer form on the growth and physiological traits, the significance (95% probability level) of the difference between N forms was directly presented in the relevant figures, and meanwhile the values of LSD_.05_ were also shown in the figures for comparing the different significance between barley genotypes. The values were averaged over four replicates.

## 5. Conclusions

The response to different N forms varies with barley genotypes. In terms of biomass, urea is a best N source for all four barley genotypes, and XZ179 was more reduced in ammonium treatment relative to other three N treatments. For tillers per plant, the two wild barley accessions were fewer in ammonium treatment than that in nitrate treatment, while the two cultivated barley genotypes had little difference between ammonium and nitrate N forms, and the most tillers in urea treatment. Root and shoot N concentrations also showed different responses to N forms among the examined four genotypes, with ZD9 containing significantly higher N concentration in nitrate and glycine treatments than in other two N treatments, and other three barley genotypes showing no difference among the four N treatments. Finally, the total protein concentration in both root and shoot differed significantly among four genotypes and form N forms, indicating the distinct effect of N metabolism by genotype and N form. It is interesting to further explore the mechanisms underlying the wild barley in the genotypic difference of responses to N forms.

## Figures and Tables

**Figure 1 plants-10-00595-f001:**
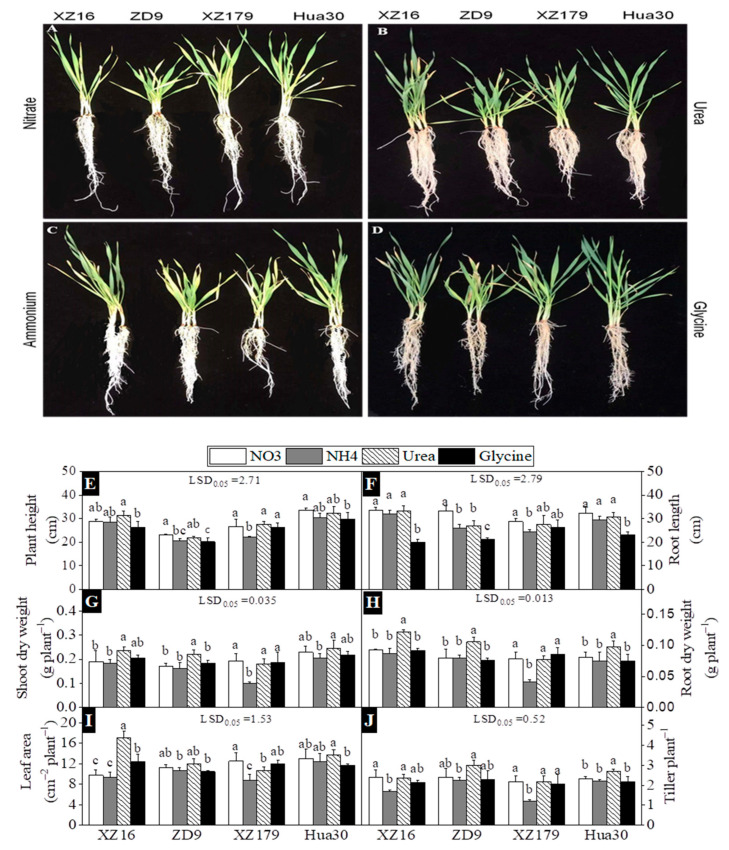
Effects of different nitrogen forms on plant morphology and growth parameters of four barley genotypes. (**A**–**D**) plant morphology after 30d of treatments (stage 4 and day 44 on Biologische Bundesanstalt, Bundessortenamt and Chemical industry scale BBCH; (**E**–**J**) are plant growth parameters. Values are expressed as mean ± SD (*n* = 4). Significant difference (*p* < 0.05) between treatments is indicated by the different letters. LSD.05 values in the figures are for the comparison between genotypes.

**Figure 2 plants-10-00595-f002:**
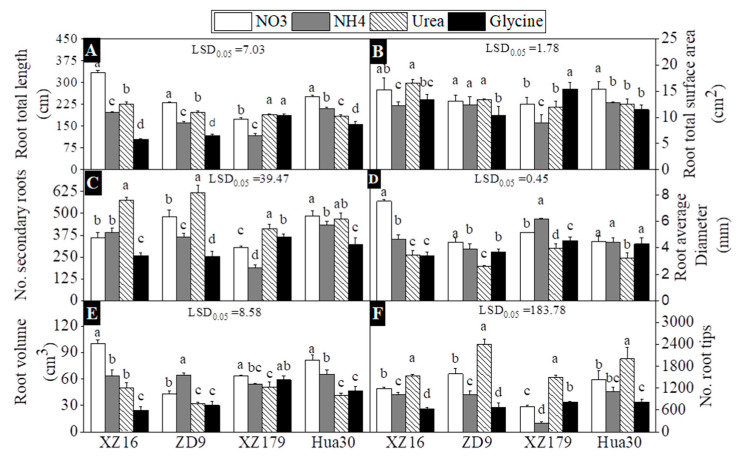
Effects of different nitrogen forms on root growth parameters Total root length (**A**)**,** total root surface area (**B**), numbers of secondary roots (**C**), Root average diameter (**D**), Root volume (**E**), and Numbers of root tips (**F**) of four barley genotypes at 28 days after treatment (stage 4 and day 42 on BBCH scale). Values are expressed as mean ± SD (*n* = 4). Significant difference (*p* < 0.05) between treatments is indicated by different letters. LSD.05 values in the figures are for the comparison between genotypes.

**Figure 3 plants-10-00595-f003:**
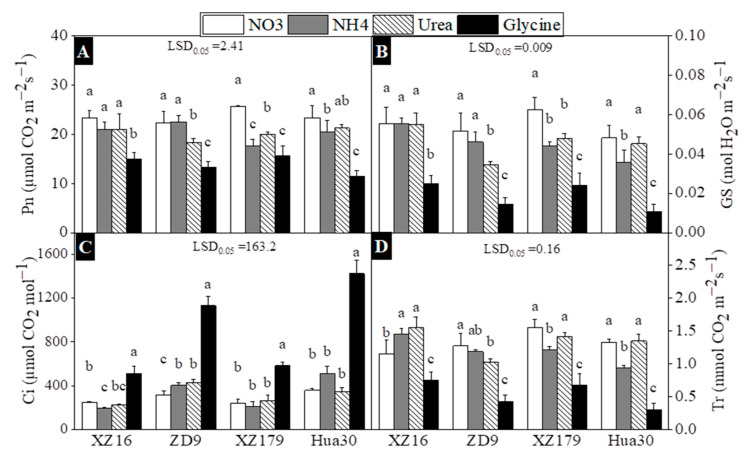
Effects of different nitrogen forms on CO_2_ assimilation rate (Pn) (**A**), stomatal conductance (Gs) (**B**), intercellular CO2 (Ci) (**C**) and transpiration rate (Tr) (**D**) of four barley genotypes at 28 day after treatment (stage 4 and day 42 on BBCH scale). Values are expressed as mean ± SD (*n* = 4). Significant difference (*p* < 0.05) between treatments is indicated by different letters. LSD.05 values in the figures are for the comparison between genotypes.

**Figure 4 plants-10-00595-f004:**
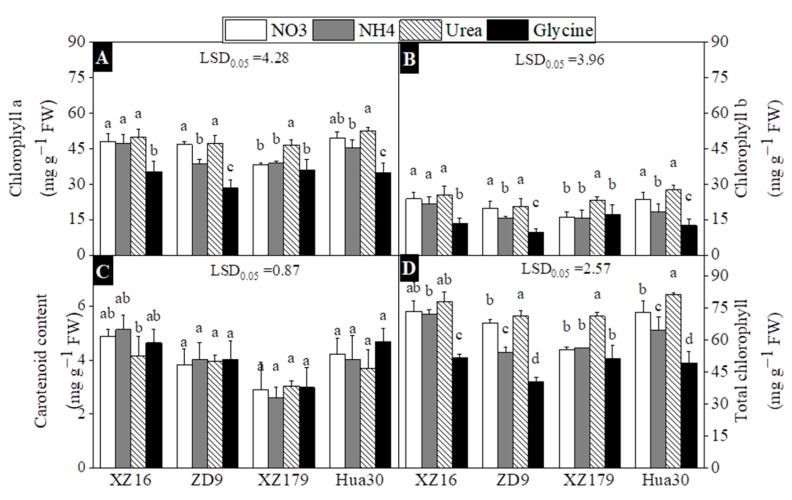
Effects of different nitrogen forms on chlorophyll a (**A**), Chlorophyll b (**B**), carotenoids (**C**), and Total chlorophyll (**D**)content of four barley genotypes at 29 day of treatment (stage 4 and day 43 on BBCH scale). All values are expressed as mean ± SD (*n* = 4). Significant difference (*p* < 0.05) between treatments is indicated by different letters. LSD.05 values in the figures are for the comparison between genotypes.

**Figure 5 plants-10-00595-f005:**
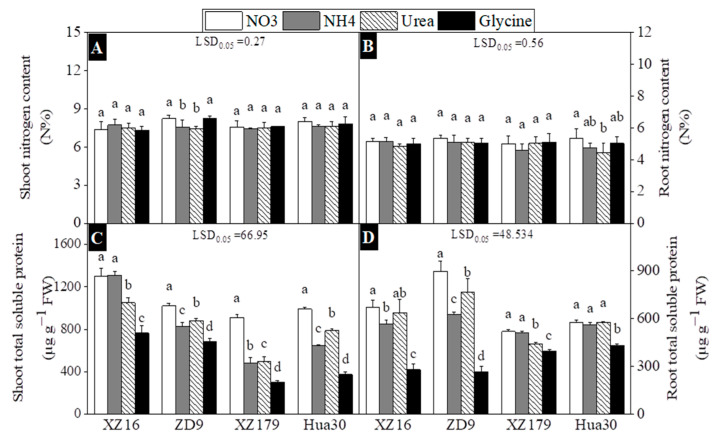
Effects of different nitrogen forms on nitrogen content (**A**,**B**) at day 30 after treatment (stage 4 and day 44 on BBCH scale) and total soluble protein contents (**C**,**D**) at day 29 after treatment (stage 4 and day 43 on BBCH scale) of four barley genotypes. Values are expressed as mean ± SD (*n* = 4). Significant difference (*p* < 0.05) between treatments is indicated by different letters. LSD.05 values in the figures are for the comparison between genotypes.

## Data Availability

Not applicable.

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
