# Peer review of "Genotypic Difference in the Responses to Nitrogen Fertilizer Form in Tibetan Wild and Cultivated Barley"

_plants, 2021, doi:10.3390/plants10030595_

Round 1
Reviewer 1 Report
In this article, authors investigated the genotypic difference of four barley genotypes in response to four N sources. Authors investigated the morphological, physiological, photosynthetic, and biochemical parameters.
Authors tend to compare between the genotypes for their results, but their current form of two ways ANOVA does not allow them to compare between genotypes. It would be better to re-do the statistical analysis that could allow them to compare between genotypes and different forms of N as well. It has to be for all the Figures.
Writing of results should be improved (especially after statistical re-analysis) and authors should cite all the figures in the text. If it is not important to cite in the text, then it can be moved to supplementary results. For examples, Authors did not comment on Figure 1 I in the text; similarly Figure 2B, 2E and 2F; and so on. Authors should check for all the figures.
It is better to mention the age of plants shown in Figure 1 A-D in the figure legends.
All the figures look quite heavy. It would be better to keep the y axis on the left for all the graphs to have two panels. For examples, one panel could be Figure 1 E, G, I , while Figure 1F,H and J could be another panel.
In material methods, it is better to mention the age of plants. For examples, section 4.3 4.4 and 4.5, there is no indication when it was performed?
Should clarify abbreviations in the text. It is better to introduce full name followed by abbreviations and then next time it is Ok to use abbreviation. Authors should check for such abbreviations. For examples TSP.
Conclusions section should be improved. In the current form, it is quite confusing for the reader to understand what is the take home message of this article.
Author Response
Reviewer (Re): Authors tend to compare between the genotypes for their results, but their current form of two ways ANOVA does not allow them to compare between genotypes. It would be better to re-do the statistical analysis that could allow them to compare between genotypes and different forms of N as well. It has to be for all the Figures.
(Au): Thanks for the comment. We made the statistical analysis according to the experimental design, the combination of two factors, genotype and N form. As this study was focused on the difference among the four N forms for each genotype, we marked the different significance between the N forms. Now we presented the values of LSD0.05 for the different comparison between the genotypes in the figures.
(Re): Writing of results should be improved (especially after statistical re-analysis) and authors should cite all the figures in the text. If it is not important to cite in the text, then it can be moved to supplementary results. For examples, Authors did not comment on Figure 1 I in the text; similarly Figure 2B, 2E and 2F; and so on. Authors should check for all the figures.
(Au): Thanks for your careful reviewing, and we cited all the figures in the suitable places.
(Re): It is better to mention the age of plants shown in Figure 1 A-D in the figure legends.
(Au): Thanks. Yes we made the changes.
(Re): All the figures look quite heavy. It would be better to keep the y axis on the left for all the graphs to have two panels. For examples, one panel could be Figure 1 E, G, I, while Figure 1F, H and J could be another panel.
(Au): Thanks. We consider the Y axis in left and light for the figures which contain two panel is easy to read for readers
(Re): In material methods, it is better to mention the age of plants. For examples, section 4.3 4.4 and 4.5, there is no indication when it was performed?
(Au): Thanks. We mentioned the age of plant for each experiment.
Re): Should clarify abbreviations in the text. It is better to introduce full name followed by abbreviations and then next time it is Ok to use abbreviation. Authors should check for such abbreviations. For examples TSP.
(Au): Thanks. We made the change.
Re): Conclusions section should be improved. In the current form, it is quite confusing for the reader to understand what is the take home message of this article.
(Au): Thanks. We changed our paper conclusion.
Reviewer 2 Report
The reviewed manuscript concerns a very important issue related to the reaction of barley to fertilization with various forms of nitrogen. Of course, it should be noted that research on the effect of nitrogen fertilization on crops is one of the most frequently studied issues. However, comparing the response of wild plants with cultivated forms is a new and very interesting issue, because wild forms may contain interesting genes that may be used in biotechnological treatments to improve barley forms. I consider conducting a comparative study on the influence of various forms of nitrogen on the growth and development of four barley genotypes, including two wild and two cultivated forms, as correct and correct both from a scientific and substantive point of view. Need to know that nitrogen plays fundamental roles in plant growth and development, as it is a necessary component of many biological macromolecules, including proteins, nucleic acids and hormones. Plants can use many kinds of N forms, ranging from inorganic N such as NH4+ and NO3− to polymeric N such as proteins. The uptake of different organic and inorganic N forms varies with plant species and availability of various N forms in soils. The authors used correct research methods, and in their analyzes they paid particular attention to: - plant growth conditions and experimental design, - measurement of morphological parameters, - gas exchange analysis, - chlorophyll content determination, - analysis of nitrogen concentration and total soluble protein. Of course, the authors subjected the obtained results to detailed statistical analysis. The quoted references in the number of 53 items is sufficient and fully illustrates the current knowledge in this field. Interesting photos and clear figures are noteworthy, as they show the different reactions of the tested barley forms to the applied doses of nitrogen compounds. The obtained results do not raise any scientific or substantive reservations. As a result of the research, the authors obtained interesting results that clearly indicate that the response to different N forms varies with barley genotypes. In terms of biomass, urea is a best N source for all four barley genotypes, and XZ179 was much reduced in ammonium treatment relative to other three N treatments. For tillers per plant, the two wild barley accessions were fewer in ammonium than in nitrate treatment, while the two cultivated barley genotypes had little difference in the two inorganic N forms. Root and shoot N concentrations also showed different responses to N forms among the examined four genotypes.
The manuscript under review should be published without changes.
Author Response
Thank you very much for your positive comment and great support.
Reviewer 3 Report
Manuscript ID: plants-1122549
Type of manuscript: Article
Title: Genotypic Difference in the Responses to Nitrogen Fertilizer Form in Tibetan Wild and Cultivated Barley
REVIEW
The paper is clearly written, easy to read, with relevant illustrative and graphical data presented. The study aimed at identifying the differences between four genotypes of barley in response to four forms of nitrogen fertilizers. As the final N concentration of the four N treatments in the nutrient solution was 2 mM and other nutrient concentrations were the same for all four treatments, no control treatment was required. The paper should really be of interest for the crop wild relative researchers and other specialists in the field. I would suggest only some minor corrections before publishing it.
– Page 6, line 1: Correct the caption by adding "carotenoid" as follows: "Figure 4. Effects of different nitrogen forms on chlorophyll and carotenoid contents…".
– Table S1 should be listed in Supplementary Materials at the end of the paper.
– In References, all entries should be formatted according to the requirements, e.g., journal title in italics, year in bold, volume number in italics, etc.
– (Optional) Bar charts illustrating closely related properties, such as chlorophyll a and b contents (Fig. 4), shoot and root N contents (Fig. 5), shoot and root total soluble protein contents (Fig. 5), should use the same vertical scale, as it was nicely done with plant height and root length presentation in Fig. 1. This would facilitate comparison of the related features and readability of the charts.
Author Response
(Re): The paper is clearly written, easy to read, with relevant illustrative and graphical data presented. The study aimed at identifying the differences between four genotypes of barley in response to four forms of nitrogen fertilizers. As the final N concentration of the four N treatments in the nutrient solution was 2 mM and other nutrient concentrations were the same for all four treatments, no control treatment was required. The paper should really be of interest for the crop wild relative researchers and other specialists in the field. I would suggest only some minor corrections before publishing it.
(Au): Thank you very much for your positive comment
(Re): Page 6, line 1: Correct the caption by adding "carotenoid" as follows: "Figure 4. Effects of different nitrogen forms on chlorophyll and carotenoid contents…".
(Au): Thanks for your valuable comment. We changed the caption by adding the carotenoids in Figure.4.
(Re): Table S1 should be listed in Supplementary Materials at the end of the paper.
(Au): Thanks. We put it in the end of the paper.
(Re): In References, all entries should be formatted according to the requirements, e.g., journal title in italics, year in bold, volume number in italics, etc.
(Au): Thanks. We changed the reference style according to the Journal requirement.
(Re): (Optional) Bar charts illustrating closely related properties, such as chlorophyll a and b contents (Fig. 4), shoot and root N contents (Fig. 5), shoot and root total soluble protein contents (Fig. 5), should use the same vertical scale, as it was nicely done with plant height and root length presentation in Fig. 1. This would facilitate comparison of the related features and readability of the charts.
(Au): Thanks for careful reviewing.
Round 2
Reviewer 1 Report
Authors answer all the queries.
However, more details should be added for statistical analysis in material and methods, like how they check distribution if data and homogeneity of variance etc. More details would help reader to understand the analysis part.
Author Response
Thank you very much for your suggestion. Accordingly we added a sentence to illustrate the presentation of statistical results in the figures, which might be helpful for readers to better understand the figures (results).